# Psoas Muscle Index as an Independent Predictor of Survival in Patients with Hepatocellular Carcinoma Receiving Systemic Targeted Therapy

**DOI:** 10.3390/cancers17020209

**Published:** 2025-01-10

**Authors:** Kenji Imai, Koji Takai, Masashi Aiba, Shinji Unome, Takao Miwa, Tatsunori Hanai, Hiroyasu Sakai, Yohei Shirakami, Atsushi Suetsugu, Masahito Shimizu

**Affiliations:** Department of Gastroenterology and internal Medicine, Graduate School of Medicine, Gifu University, 1-1 Yanagido, Gifu 501-1194, Japan; takai.koji.t2@f.gifu-u.ac.jp (K.T.); aiba.masashi.v4@f.gifu-u.ac.jp (M.A.); unome.shinji.t7@f.gifu-u.ac.jp (S.U.); hanai.tatsunori.p8@f.gifu-u.ac.jp (T.H.); sakai.hiroyasu.a8@f.gifu-u.ac.jp (H.S.); shirakami.yohei.t3@f.gifu-u.ac.jp (Y.S.); suetsugu.atsushi.e2@f.gifu-u.ac.jp (A.S.); shimizu.masahito.j1@f.gifu-u.ac.jp (M.S.)

**Keywords:** hepatocellular carcinoma, targeted therapy, sarcopenia, psoas muscle index, overall survival

## Abstract

The psoas muscle index (PMI) is a more practical and accessible maker of skeletal muscle mass compared to the skeletal muscle index (SMI), as it can be measured solely by the length of the psoas muscle. In patients with advanced hepatocellular carcinoma undergoing targeted therapy, the PMI demonstrated a strong correlation with the SMI across both sexes. Furthermore, the optimal cut-off values identified through the maximally selected statistics (2.86 cm^2^/m^2^ for women and 3.55 cm^2^/m^2^ for men) significantly stratified patient outcomes. Consequently, the PMI serves as a reliable surrogate for the SMI in assessing skeletal muscle mass and predicting survival following systemic targeted therapy.

## 1. Introduction

Hepatocellular carcinoma (HCC), which typically develops in cirrhotic livers, is one of the most common malignancies worldwide [1]. Tumor progression and liver function reserves are associated with reduced survival in patients with HCC [2,3]. For example, the levels of α-fetoprotein (AFP), a tumor marker for HCC, and the ALBI score, an indicator of liver function reserve, define the prognosis of HCC [3,4]. Patients with HCC often have sarcopenia, which is characterized by the degenerative loss of skeletal muscle mass, quality, and strength [5]. Sarcopenia has also been reported to predict prognosis in these patients [6]. Effective management strategies, including nutritional interventions such as tailored dietary counseling, branched-chain amino acid (BCAA) and levocarnitine supplementation, and exercise therapy, are recommended to mitigate sarcopenia in patients with HCC [7,8]. To ensure the timely implementation of these interventions, the continuous assessment of skeletal muscle mass is essential. Consequently, there is an increasing need for a simpler and more reliable method to evaluate skeletal muscle mass.

Sarcopenia guidelines recommend the use of bioelectrical impedance analysis and computed tomography (CT) imaging to assess skeletal muscle mass [9]. The skeletal muscle index (SMI) is calculated by dividing the cross-sectional area of the skeletal muscle at the third lumbar vertebra level on CT images by the square of the individual’s height. This measurement can be obtained from CT images commonly used to assess HCC. The SMI has been reported to be a useful indicator for prognosis assessment and pathology in patients with HCC [6]. Therefore, the SMI is now considered the gold standard for assessing skeletal muscle mass in patients with liver disease; however, expensive image analysis software is required to measure the SMI, and CT scans have radiation exposure issues.

The psoas muscle index (PMI) is derived by dividing the sum of the products of the major and minor axes of the left and right psoas muscles at the level of the third lumbar vertebra by the square of the individual’s height. The PMI has an advantage over the SMI in that it only requires the measurement of the psoas muscle’s length, eliminating the need for expensive image analysis software, and it can be assessed using magnetic resonance imaging (MRI). In addition, the PMI is highly correlated with the SMI [9]. Therefore, the PMI is recognized as a viable alternative to the SMI when SMI measurements are not feasible [9]. However, the utility of the PMI in clinical practice for predicting survival remains unexplored. In particular, it has not been adequately tested whether the cut-off values for the PMI used in sarcopenia guidelines (6.0 cm^2^/m^2^ for men and 3.4 cm^2^/m^2^ for women) [9] are useful for predicting survival in these patients.

The primary aim of this study was to clarify the relationship between the PMI and the SMI in patients with HCC receiving systemic targeted therapy. The secondary aim was to determine whether the PMI, in addition to established prognostic factors such as liver function reserve and tumor characteristics, can serve as an independent predictor of survival after targeted therapy. Moreover, we sought to identify the optimal PMI cut-off value that yielded the most significant survival.

## 2. Materials and Methods

### 2.1. Patients and Treatment Strategy

A total of 214 patients with HCC who received systemic targeted therapy, including atezolizumab plus bevacizumab (*n* = 43), tremelimumab plus durvalumab (*n* = 5), lenvatinib (*n* = 76), sorafenib (*n* = 79), cabozantinib (*n* = 6), ramcirumab (*n* = 3), and regorafenib (*n* = 1), from May 2009 to December 2023 at the Gifu University Hospital were enrolled. The patient demographics and clinical details are summarized in Table 1. A total of 19 patients had no previous treatment history for HCC, whereas 195 patients had undergone some form of previous treatment, including 52 patients who had received other systemic therapies. The drug selection was based on the latest systemic targeted therapy guidelines for HCC [10]. Dynamic CT or MRI was performed at the start of systemic targeted therapy, and it was subsequently deemed necessary until death or the end of the observation period by the treating physician. Treatment response was assessed using the modified RECIST criteria [11]. Overall survival (OS) was defined as the interval from the start of systemic targeted therapy until death or the end of the observation period in December 2023.

### 2.2. Methods for Calculating the Skeletal Muscle Index and the Psoas Muscle Index

The methodology for calculating the SMI and the PMI is shown in Figure 1. The SMI was derived by dividing the cross-sectional area of skeletal muscle at the level of the third lumbar vertebra on CT images (the green area in Figure 1) by the square of the individual’s height, using SYNAPSE VINCENT software (version 6.7, Fujifilm Medical, Tokyo, Japan). The PMI was calculated by dividing the sum of the products of the major and minor axes of the left and right psoas muscles at the level of the third lumbar vertebra on CT or MRI images (A × B + C × D in Figure 1) by the square of the individual’s height. Throughout this study, the SMI was measured 531 times, with an average of 2.5 measurements per individual, while the PMI was measured 635 times, with an average of 3.0 measurements per individual. Both the SMI and the PMI were measured by a single gastroenterologist with more than 20 years of clinical experience.

### 2.3. Statistical Analyses

The correlation between the PMI and the SMI was assessed using Pearson’s correlation coefficient (PCC) based on 531 concurrent PMI and SMI measurements. Survival was estimated employing the Kaplan–Meier method, and differences between survival curves were evaluated with the log-rank test. To determine whether the PMI independently influenced survival after the initiation of systemic targeted therapy, it was analyzed alongside established prognostic factors, including the AFP and the ALBI score [2,3], using the Cox proportional hazard model. Given the multiple measurements per individual in this study, the PMI, the AFP, and the ALBI score were treated as time-dependent covariates [12]. The AFP and the ALBI score obtained on or around the date of PMI measurement were used in the Cox proportional hazard model. The optimal PMI cut-off value that yielded the most significant differences in survival was determined separately for men and women using maximally selected statistics [13]. The baseline characteristics were compared using the Mann–Whitney U test for continuous variables or Fisher’s exact test for categorical variables. A *p*-value < 0.05 was considered statistically significant. Analyses were performed using the R software ver. 4.4.1, with the packages ‘survival (ver. 3.8.3)’, ‘Rcmdr (ver. 2.9.5)’, ‘maxstat (ver. 0.7.25)’, and ‘ggplot2 (ver. 3.5.1)’ (R Foundation for Statistical Computing, Vienna, Austria; http://www.R-project.org/ (accessed on 22 August 2024)).

Among the 214 patients, 7 achieved a complete response following conversion therapy, 93 transitioned to subsequent treatment, and 96 could not proceed to subsequent treatment due to deterioration in their overall condition or other factors. The remaining 18 patients continued to receive systemic treatment until the end of the observation period. The patient flowchart for this study is shown in Appendix A.

The patients enrolled in this study were given the opportunity to opt out, with a full disclosure of the study details. The study design, including the consent procedure, was approved by the Ethics Committee of the Gifu University School of Medicine on 12 December 2023 (ethical protocol code: 29–26).

## 3. Results

### 3.1. Baseline Clinical Characteristics and Treatment Response of the Enrolled Patients

This study included 171 men and 43 women (Table 1). The median age, SMI, PMI, ALBI score, and AFP level at the baseline were 73 years, 42.96 cm^2^/m^2^, 4.59 cm^2^/m^2^, −2.400, and 0.070 × 10^3^ ng/mL, respectively. The median treatment duration was 4.9 months. Complete response, partial response, stable disease, and progressive disease were observed in 15 (7.0%), 32 (15.0%), 78 (36.4%), and 82 (38.3%) patients, respectively; 7 patients were not evaluable. The objective response and disease control rates were 22.7% and 60.4%, respectively. The OS rates after 1, 2, and 3 years and the median OS were 62.9%, 35.5%, 19.8%, and 17.2 months, respectively. In total, 132 deaths occurred during the study period.

### 3.2. Correlation Between PMI and SMI in Patients with HCC Undergoing Systemic Targeted Therapy

The PCC between the PMI and the SMI was 0.38 for women (*p* < 0.001; *n* = 100) and 0.62 for men (*p* < 0.001; *n* = 431; Figure 2). These results indicated a significant correlation between the PMI and the SMI in both women and men.

### 3.3. Impact of PMI and SMI on Survival in Patients with HCC Undergoing Systemic Targeted Therapy

A multivariate analysis was performed to identify the factors affecting survival after the initiation of systemic targeted therapy (Table 2) using the Cox proportional hazard model. The PMI, SMI, AFP, and ALBI score were treated as time-dependent covariates. The analysis showed that the PMI (hazard ratio [HR]: 0.852; 95% confidence interval [CI]: 0.755–0.962; and *p* < 0.001) and the SMI (HR: 0.951; 95% CI: 0.932–0.971; and *p* < 0.001) independently influenced patient survival even after adjusting for known prognostic factors, the AFP, and the ALBI score. Furthermore, even when the analysis was restricted to the cohort of patients receiving lenvatinib or sorafenib, the PMI remained an independent prognostic factor (Appendix A).

### 3.4. Optimal PMI Cut-Off Value for Survival in Patients with HCC Undergoing Systemic Targeted Therapy

The maximally selected statistics showed that the optimal PMI cut-off values that yielded the most significant differences in survival were 2.86 cm^2^/m^2^ for women (Figure 3a) and 3.55 cm^2^/m^2^ for men (Figure 3b). Using these cut-off values, the initial number of patients classified within the lower-PMI group was 32 (7 women and 25 men), which increased to 72 (15 women and 57 men) during the observation period. As indicated in Appendix A, which compares the clinical characteristics at the baseline and the adverse events between the lower- and higher-PMI groups, hand–foot syndrome as a response to systemic therapy was the sole significant differentiator between these groups. Moreover, patients in the lower-PMI group (the OS rates after 1 year and the median OS were 47.3% and 3.2 months for women and 53.6 % and 13.1 months for men) had a significantly shorter survival time compared to those in the higher-PMI group (the OS rates after 1 year and the median OS were 69.7% and unachieved for women and 73.2% and unachieved for men, as shown in Figure 3c,d). Furthermore, these PMI cut-off values demonstrated effectiveness in stratifying survival, even when the analysis was restricted to the cohort of patients receiving sorafenib or lenvatinib (Appendix A).

## 4. Discussion

The present study demonstrated that the PMI is significantly correlated with the SMI in both women and men with unresectable advanced HCC, which is consistent with previous studies [9]. Furthermore, in addition to established prognostic factors such as the AFP and the ALBI score, the PMI independently influenced survival after the initiation of systemic targeted therapy. To the best of our knowledge, this is the first study to focus on patients with advanced HCC undergoing systemic targeted therapy, consider the PMI a time-dependent covariate, and establish the PMI as an independent prognostic factor. These results validate the PMI as a reliable surrogate for the SMI for assessing skeletal muscle mass and predicting prognosis, as suggested by the sarcopenia guidelines [9]. The PMI can be determined using only linear measurements, thereby obviating the need for expensive image analysis software. Furthermore, it can be measured not only on CT scans but also on MRI scans, making it superior to the SMI. The use of the PMI to assess skeletal muscle mass over time and predict prognosis in patients with HCC is particularly important in institutions where SMI measurements are unavailable.

The optimal PMI cut-off values that yielded the most significant differences in survival (2.86 cm^2^/m^2^ for women and 3.55 cm^2^/m^2^ for men) were identified in the present study using maximally selected statistics. This study is the first to propose PMI cut-off values for patients with HCC undergoing systemic targeted therapy. Identifying the most appropriate PMI cut-off value for prognostic prediction is of great value for the routine clinical management of patients with HCC. Interestingly, the cut-off values identified in this study were much lower than those recommended by the sarcopenia guidelines in Japan (3.4 cm^2^/m^2^ for women and 6.0 cm^2^/m^2^ for men) [9]. Furthermore, when applying the latter cut-off value to our cohort, 136 patients (63.6%) were classified into the sarcopenia group at the start of the observation period, with no significant differences in survival after systemic targeted therapy. These facts imply that initiating sarcopenia management based on the existing sarcopenia criteria (3.4 cm^2^/m^2^ for women and 6.0 cm^2^/m^2^ for men) may be too late for patients with HCC undergoing systemic targeted therapy. Patients with HCC undergoing systemic targeted therapy should be rigorously assessed for skeletal muscle mass loss before being diagnosed with overt sarcopenia, and appropriate measures should be taken. More attention may be needed in this regard, especially in men, owing to the large differences in the cut-off values (3.55 cm^2^/m^2^ vs. 6.0 cm^2^/m^2^).

Skeletal muscle mass varies greatly depending on the stage of liver disease. We have previously shown that skeletal muscle mass decreases exponentially as the liver function reserve deteriorates and HCC progresses [14,15]. In addition, treatment of HCC with sorafenib or lenvatinib has been shown to significantly reduce skeletal muscle mass, whereas atezolizumab plus bevacizumab treatment has not [15,16]. These results suggest that the skeletal muscle mass in patients with HCC following the initiation of systemic therapy is influenced by factors such as the type of agent used, the therapeutic efficacy, the progression of liver function reserve, and subsequent treatments. In fact, the number of patients classified into the lower-PMI group (≤2.86 cm^2^/m^2^ for women and ≤3.55 cm^2^/m^2^ for men) in this study increased from 32 to 72 over the course of the observation period. Therefore, when assessing skeletal muscle mass in patients with HCC undergoing systemic targeted therapy, it is essential to perform repeated measurements rather than relying solely on baseline assessments. Furthermore, this study, which involved multiple PMI measurements in the same patient while treating the PMI as a time-dependent covariate, unequivocally demonstrated that the PMI facilitates the accurate assessment of temporal changes in skeletal muscle mass. Furthermore, PMI measurements obtained at various time points serve as significant predictors of survival following systemic targeted therapy. Generally, lenvatinib surpasses sorafenib in terms of progression-free survival, objective response, and disease control rate; however, it is not associated with a significant extension of the OS [17]. This discrepancy can be explained by the impact of subsequent therapies administered after disease progression. Lenvatinib and sorafenib were the most frequently administered agents in this study. When the analysis was restricted to either agent, the PMI emerged as an independent prognostic factor, and the proposed PMI cut-off value proved effective for prognostic stratification. Therefore, evaluating skeletal muscle mass even after the initiation of subsequent therapy using the PMI and initiating sarcopenia interventions, such as nutritional and exercise therapies, based on our identified cut-off values (≤2.86 cm^2^/m^2^ for women and ≤3.55 cm^2^/m^2^ for men) appears highly appropriate for improving patient prognosis.

Regarding nutritional interventions for addressing sarcopenia in patients with chronic liver diseases, including HCC, several studies have demonstrated the effectiveness of nutritional counseling, regular monthly consultations with dietitians, and late-evening snacks enriched with carbohydrates or proteins [18,19]. Supplementation with BCAA and/or levocarnitine has also shown promise as an intervention [7,8]. Additionally, walking for 30 min three times per week and resistance exercises have been shown to improve muscle mass in cirrhotic patients [20]. Furthermore, a study reported that combining oral BCAA supplementation with exercise therapy effectively prevents a decline in skeletal muscle mass [21]. It is essential to recognize that patients with HCC, particularly those undergoing systemic therapy, are at heightened risk of significant reductions in skeletal muscle mass. Therefore, the sarcopenia interventions described above should be implemented proactively before the condition becomes difficult to manage.

This study had certain limitations. This was a retrospective, single-center study with a relatively small sample size. The PMI cut-off values identified in this study are specific to Japanese patients with HCC undergoing systemic targeted therapy and may not be generalizable to all patients with liver disease because the skeletal muscle mass varies greatly depending on the stage of liver disease and race [14,22,23]. Moreover, this study focused solely on skeletal muscle mass, without assessing muscle strength, which is essential for the diagnosis of sarcopenia [9,24]. This retrospective study included many patients without grip strength measurements, precluding the evaluation of the prognostic impact of muscle strength. This study included cases where systemic therapy was administered as both first-line and later-line treatments. Notably, in cases limited to later-line therapy, the prognostic stratification based on the PMI was less pronounced. The frequency of PMI measurements and the intervals between them varied considerably among cases, leading to numerous survival data points with very short observation periods. Although the PMI is determined solely based on the linear distance of the psoas muscle, the possibility of intra-observer variability cannot be entirely excluded. The slightly weaker correlation between the PMI and the SMI observed particularly in women, as well as the attenuated prognostic stratification based on the PMI in later-line cases, may be attributable to the factors mentioned above. Prospective studies with larger cohorts of patients, including those of different races and liver disease stages, are required to address these issues.

## 5. Conclusions

The PMI significantly correlated with the SMI and independently influenced the survival of patients with HCC following systemic targeted therapy. The optimal cut-off values identified through the maximally selected statistics (2.86 cm^2^/m^2^ for women and 3.55 cm^2^/m^2^ for men) significantly stratified the patient outcomes, although these values may require revision based on findings from future prospective studies. Consequently, the PMI serves as a reliable surrogate for the SMI in assessing skeletal muscle mass and predicting survival.

## Figures and Tables

**Figure 1 cancers-17-00209-f001:**
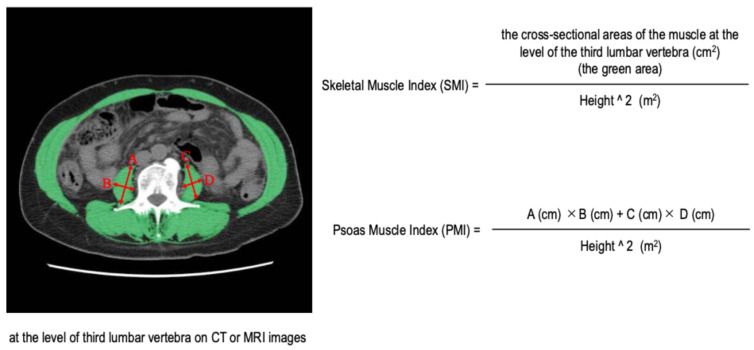
Methods for calculating the skeletal muscle index and the psoas muscle index.

**Figure 2 cancers-17-00209-f002:**
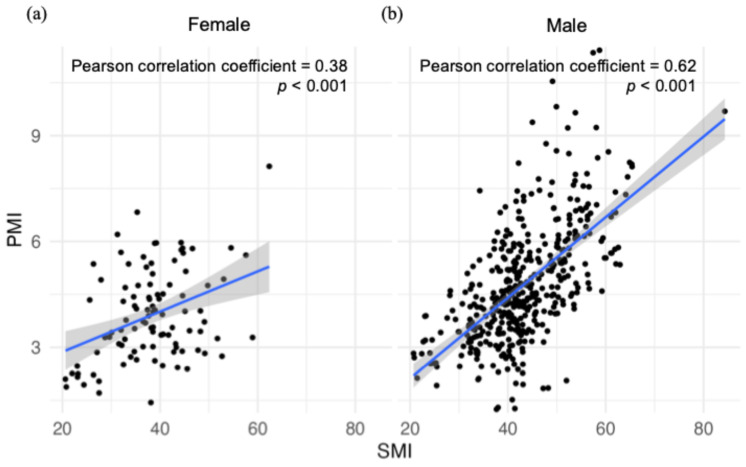
Pearson’s correlation coefficients between PMI and SMI for (**a**) women and (**b**) men. The blue line represents the least-squares regression line, while the gray-shaded area indicates the 95% confidence interval.

**Figure 3 cancers-17-00209-f003:**
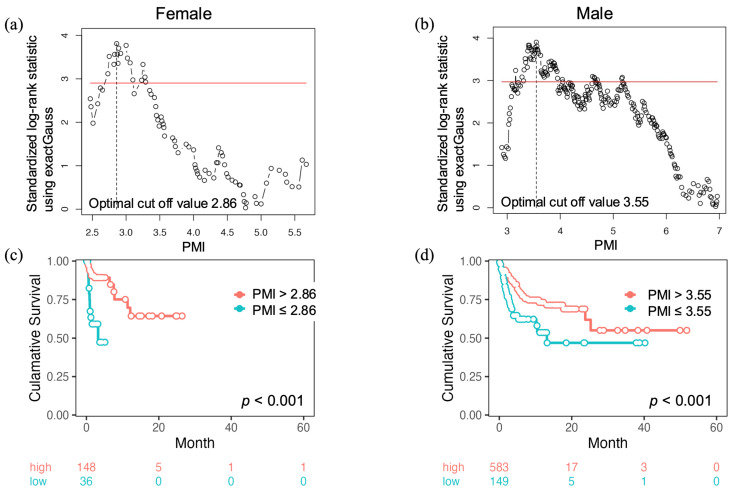
Optimal PMI cut-off values yielding the most significant differences in survival following systemic targeted therapy. These values were determined using maximally selected statistics for (**a**) women and (**b**) men. Kaplan–Meier curves for cumulative survival after systemic targeted therapy initiation divided by the optimal PMI cut-off value for (**c**) women (>2.86 and ≤2.86 cm^2^/m^2^) and (**d**) men (>3.55 and ≤3.55 cm^2^/m^2^).

**Table 1 cancers-17-00209-t001:** Baseline demographic and clinical characteristics of the enrolled patients.

Variables	
Age (years)	73 [67, 79]
Sex (man/woman)	171/43
Etiology (HBV/HCV/others)	44/63/107
Drug (AB/DT/LEN/SOR/CAB/RAM/REG)	43/5/76/79/6/3/1
BCLC stage (A/B/C)	13/82/116
Child (A/B)	190/24
ALBI score	−2.400 [−2.755, −2.080]
ECOG PS (0/1/2/3)	162/39/12/1
BCAA supplementation (yes/no)	71/143
Levocarnitine supplementation (yes/no)	26/188
Previous treatment (no/resection/RFA/TACE/RT/other systemic therapy)	19/99/63/148/50/52
AFP (×10^3^ ng/mL)	0.070 [0.006, 1.521]
PIVKA-II (×10^3^ mAU/mL)	0.630 [0.073, 4.104]
SMI (cm^2^/m^2^)	42.96 [38.74, 49.10]
SMI measurement frequency (total/average)	531/2.5
PMI (cm^2^/m^2^)	4.59 [3.77, 5.74]
PMI measurement frequency (total/average)	635/3.0
Best response (CR/PR/SD/PD/NE)	15/32/78/82/7
Treatment duration (month)	4.9 [1.7, 11.1]

Continuous covariates are presented as medians [interquartile range]. HBV, hepatitis B virus; HCV, hepatitis C virus; AB, atezolizumab/bevacizumab; DT, tremelimumab/durvalumab; LEN, lenvatinib; SOR, sorafenib; CAB, cabozantinib; RAM, ramucirumab; REG, regorafenib; BCLC stage, Barcelona Clinic Liver Cancer stage; ECOG, Eastern cooperative oncology group; PS, performance status; BCAA, branched-chain amino acid; RFA, radiofrequency ablation; TACE, transcatheter arterial chemoembolization; RT, radiation therapy; AFP, alpha-fetoprotein; PIVKA-II, protein induced by vitamin K absence or antagonist-II; SMI, skeletal muscle index; PMI, psoas muscle index; CR, complete response; PR, partial response; SD, stable disease; and PD, progressive disease.

**Table 2 cancers-17-00209-t002:** Univariate and multivariate analyses of predictors for survival using the time-dependent Cox proportional hazard model.

	HR (95%CI)	*p*-Value
Univariate analysis		
PMI (cm^2^/m^2^)	0.789 (0.698–0.893)	<0.001
SMI (cm^2^/m^2^)	0.933 (0.912–0.954)	<0.001
AFP (×10^3^ ng/mL)	1.001 (1.001–1.002)	0.006
ALBI score	3.794 (2.943–4.890)	<0.001
Multivariate analysis (adjusted for AFP and ALBI score)		
PMI (cm^2^/m^2^)	0.852 (0.755–0.962)	<0.001
SMI (cm^2^/m^2^)	0.951 (0.932–0.971)	<0.001

All the variables were dealt with as time-varying covariates. HR, hazard ratio; CI, confidence interval; PMI, psoas muscle index; SMI, skeletal muscle index; AFP, alpha-fetoprotein; and ALBI score, albumin–bilirubin score.

## Data Availability

The data presented in this study are available upon request from the corresponding author.

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
