# Peer review of "Psoas Muscle Index as an Independent Predictor of Survival in Patients with Hepatocellular Carcinoma Receiving Systemic Targeted Therapy"

_cancers, 2025, doi:10.3390/cancers17020209_

Round 1

Reviewer 1 Report

Comments and Suggestions for Authors

The study is interesting but i have a few comments:

1) Could the authors perform a subgroup analysis at least in the two most frequent treatments (sorafenib and lenvatinib)?

2) Following the previous comment, the authors should comment on the state of the art in this field, particularly concerning the two aforementioned drugs (in this regard, cite the recent SRMA: PMID: 34017396)

3) Please add the number at risk in the KM curves

4) I have some concerns about the short follow-up....some groups didn't even reach the median survival.....

Author Response

Response to Reviewer #1

We are pleased that in the overall comments this reviewer found our study is of interest. We also thank this reviewer’s constructive comments which were most helpful to improve our manuscript. We accordingly revised the manuscript as follows.

  1. Could the authors perform a subgroup analysis at least in the two most frequent treatments (sorafenib and lenvatinib)?

Following your suggestion, we conducted a survival analysis focusing on cases treated with sorafenib or lenvatinib. As shown in the newly created Table S1 and Figure S2, even when limited to these cases, the proposed PMI cutoff values (2.86 cm²/m² for women and 3.55 cm²/m² for men) effectively stratified prognosis, and PMI was identified as an independent prognostic factor. These results and their discussion have been incorporated into the Results section (lines 172–173 and 193–195).

  1. Following the previous comment, the authors should comment on the state of the art in this field, particularly concerning the two aforementioned drugs (in this regard, cite the recent SRMA: PMID: 34017396)

The referenced citation highlights that lenvatinib surpasses sorafenib in terms of progression-free survival, objective response, and disease control rate; however, it is not associated with a significant extension of OS. This discrepancy can be explained by the impact of subsequent therapies administered after disease progression. Lenvatinib and sorafenib were the most frequently administered agents in this study. When the analysis was restricted to either agent, PMI emerged as an independent prognostic factor, and the proposed PMI cutoff value proved effective for prognostic stratification. Therefore, evaluating skeletal muscle mass even after the initiation of subsequent therapy using PMI and initiating sarcopenia interventions, such as nutritional and exercise therapies, based on our identified cutoff values (≤ 2.86 cm2/m2 for women and ≤ 3.55 cm2/m2 for men) appears highly appropriate for improving patient prognosis. We have incorporated the above statement into the Discussion section (lines 251–258), accompanied by a newly added reference (#17). Your insights have further substantiated the significance of our proposed PMI measurement. Thank you for your valuable contribution.

  1. Please add the number at risk in the KM curves

In accordance with this suggestion, we have added the number at risk to Figure 3.

  1. I have some concerns about the short follow-up....some groups didn't even reach the median survival.....

The frequency of PMI measurements and the intervals between them vary significantly among cases, resulting in numerous survival data points with exceedingly short observation periods. We have incorporated this fact as a limitation of the study into the Discussion section (lines 284–286). We appreciate your valuable insights in highlighting this critical issue in our manuscript.

Reviewer 2 Report

Comments and Suggestions for Authors

The authors investigated the psoas muscle index (PMI) in patients with hepatocellular carcinoma (HCC) receiving systemic therapy. They found that PMI was well correlated with the skeletal muscle index (SMI), and PMI was associated with survival after the initiation of systemic therapy. This is a well-written paper that presents interesting data. However, the manuscript still raises several issues.

1)    The cohort of 214 patients included 35 individuals who had undergone second-line or subsequent systemic therapy, such as cabozantinib, ramucirumab, and regorafenib. I wonder if it would be unreasonable to perform a survival analysis by combining first-line treatment with second-line or later treatments.  Furthermore, the authors should consider the potential confounding effect of prior therapy on the assessment of sarcopenia at the initiation of subsequent systemic therapy in patients receiving a second or subsequent systemic therapy for HCC.

2)    In Table 1, it would be better to add whether or not patients were taking BCAAs and/or levocarnitine, which are considered useful in ameliorating sarcopenia during systemic therapy for HCC (PMID: 34974449, PMID: 34959980). For the 35 patients who underwent second-line or subsequent systemic therapy, the impact of medications introduced during the first-line treatment warrants consideration.

3)    The footnote in Table 1 contains the spelling error for ramucirumab.

4)    The potential for inter-rater reliability exists in the PMI analysis. How many individuals conducted the PMI analysis?

5)    What is the etiology of the outlier observed in Figure 2?

6)    The resolution of Figure 3 (a) and (b) is suboptimal, and the font size is inadequate.

7)    The significance of sarcopenia management in conjunction with sarcopenia evaluation is addressed in the introduction section. Further discussion is required to address the sarcopenia management.

Author Response

Response to Reviewer #2

We are pleased that in the overall comments this reviewer found our study is of interest. We also thank this reviewer’s constructive comments which were most helpful to improve our manuscript. We accordingly revised the manuscript as follows.

  1. The cohort of 214 patients included 35 individuals who had undergone second-line or subsequent systemic therapy, such as cabozantinib, ramucirumab, and regorafenib. I wonder if it would be unreasonable to perform a survival analysis by combining first-line treatment with second-line or latertreatments. Furthermore, the authors should consider the potential confounding effect of prior therapy on the assessment of sarcopenia at the initiation of subsequent systemic therapy in patients receiving a second or subsequent systemic therapy for HCC.

After reviewing the data, we corrected the number of later-line cases to 52. When dividing the first-line and second-line cases using the PMI cutoff values we proposed (2.86 cm²/m² for women and 3.55 cm²/m² for men), no statistically significant differences were observed (please refer to the attached materials). Possible explanations include the small sample size, intra-observer variability, and survival data points with exceedingly short observation periods. Nonetheless, the coexistence of first-line and later-line cases constitutes a significant limitation of this study, which has been incorporated into the Discussion section (lines 282–291). We sincerely appreciate your valuable feedback in identifying this critical issue.

  1. In Table 1, it would be better to add whether or not patients were taking BCAAs and/or levocarnitine, which are considered useful in ameliorating sarcopenia during systemic therapy for HCC (PMID: 34974449, PMID: 34959980). For the 35 patients who underwent second-line or subsequent systemic therapy, the impact of medications introduced during the first-line treatment warrants consideration.

In accordance with your suggestion, we have added the items “BCAA supplementation” and “levocarnitine supplementation” to Table 1. Using the data from this study, we retrospectively compared the rates of BCAA and levocarnitine supplementation between the lower and higher PMI groups. However, no statistically significant differences were observed between the two groups at either the initiation of systemic therapy or the final observation point (please refer to the attached materials). This result is likely attributable to the tendency to administer these agents to cases with more advanced liver cirrhosis. To avoid potential misinterpretation by readers, we refrained from discussing these results in the main text. Nevertheless, recognizing the importance of these agents as interventions against sarcopenia, we have incorporated this point into the Introduction (lines 49–54) and Discussion (lines 265–266) sections along with the suggested references (#7 and #8). We sincerely appreciate your invaluable feedback.

  1. The footnote in Table 1 contains the spelling error for ramucirumab.

The points you highlighted have been duly revised (lines 100).

  1. The potential for inter-rater reliability exists in the PMI analysis. How many individuals conducted the PMI analysis?

Both SMI and PMI were measured by a single gastroenterologist with over 20 years of clinical experience (lines 124–126). Although PMI is determined solely based on the linear distance of the psoas muscle, the possibility of intra-observer variability cannot be entirely excluded. We have added this point as a limitation of our study in the Discussion section (lines 286–288). Thank you for highlighting this important aspect.

  1. What is the etiology of the outlier observed in Figure 2?

We investigated the characteristics of outliers from the perspectives of background liver condition, age, gender, medications used, liver functional reserve, and clinical stage; however, no significant correlations were identified. Therefore, we regret to inform you that we are unable to provide a definitive answer to this question.

  1. The resolution of Figure 3 (a) and (b) is suboptimal, and the font size is inadequate.

In accordance with this suggestion, we have appropriately revised Figure 3. Thank you for your valuable feedback.

  1. The significance of sarcopenia management in conjunction with sarcopenia evaluation is addressed in the introduction section. Further discussion is required to address the sarcopenia management.

In response to this suggestion, we incorporated measures against sarcopenia, focusing on nutritional and exercise therapies, by citing relevant literature (#7, #8, #18–#21) and added these details to the Introduction section (lines 49–54) and the Discussion section (lines 262–273). Notably, in the Discussion section, we emphasized that “the sarcopenia interventions described above should be implemented proactively before the condition becomes difficult to manage.”

Reviewer 3 Report

Comments and Suggestions for Authors

This very interesting study investigates the prognostic value of the Psoas Muscle Index (PMI) as an independent predictor of survival in patients with HCC undergoing systemic therapy. PMI was compared to the Skeletal Muscle Index, and its correlation with survival outcomes was assessed. The authors identified optimal PMI cut-off values (2.86 cm²/m² for women and 3.55 cm²/m² for men) using maximally selected statistics and demonstrated that PMI independently predicts survival alongside established factors like AFP and ALBI score. The study concludes that PMI serves as a more accessible and practical alternative to SMI for assessing sarcopenia and predicting survival outcomes in this patient population.

Major Issues

1) In the study, the exact anatomical point for measuring the psoas muscle to calculate the PMI is not clearly specified. This lack of clarity can introduce variability in measurements, potentially reducing the accuracy and reproducibility of the results. It is essential to establish a standardized anatomical reference point for measurement, ideally based on a specific vertebral level (e.g., the third lumbar vertebra, L3). This approach would ensure consistency across different patients and research centers, improving the validity and comparability of the findings with other studies. The authors are encouraged to explicitly define the vertebral level used for psoas muscle measurement and include this information in both the Methods section and relevant illustrative figures.

2) The study is a single-center, retrospective analysis, which limits the generalizability of the findings; moreover, the lack of external validation for the PMI cut-off values weakens their clinical applicability. Authors should underline it in the conclusion.

3) The study appropriately uses time-dependent covariates (PMI, AFP, ALBI score), but it does not address potential biases from repeated measurements adequately.

4) The reported correlation between PMI and SMI is moderate (PCC = 0.38 for women which shows a weak correlation, PCC = 0.62 for men which shows a moderate correlation), particularly weaker in women, which challenges the claim that PMI is a reliable surrogate for SMI.

5) Sarcopenia is defined by both muscle mass and muscle strength. The study evaluates only mass, neglecting an essential diagnostic parameter, which reduces the robustness of the conclusions.

6) The optimal PMI cut-off values were derived statistically but not validated externally. Without broader validation, these values may lack general applicability in diverse patient populations.

Minor Issues

1) Complex sentence structures and redundancies, particularly in the Abstract and Introduction, reduce clarity. Some phrases, like "independently influenced survival following systemic targeted therapy," are repeated excessively.

2) Figure 1: hight is not an English word, I think authors meant height as in the text. This must be corrected. Moreover, Height x Height can be represented in math as Height^2.

3) Figure 3: please annotate median survival values and key survival time points on the Kaplan-Meier curves for enhanced interpretability.

4) Key findings are unnecessarily repeated across multiple sections, reducing the manuscript's overall readability and impact.

The study addresses an important clinical question regarding the prognostic value of PMI in patients with advanced HCC undergoing systemic targeted therapy. Despite methodological rigor in statistical analysis, the lack of external validation, moderate correlation between PMI and SMI, and the absence of muscle strength assessment limit the strength of the conclusions.

With major revisions addressing the outlined issues, the manuscript has strong potential for acceptance and could significantly contribute to the field of sarcopenia research in HCC patients.

Author Response

Response to Reviewer #3

We are pleased that in the overall comments this reviewer found our study is of interest. We also thank this reviewer’s constructive comments which were most helpful to improve our manuscript. We accordingly revised the manuscript as follows.

Major Issues

  1. In the study, the exact anatomical point for measuring the psoas muscle to calculate the PMI is not clearly specified. This lack of clarity can introduce variability in measurements, potentially reducing the accuracy and reproducibility of the results. It is essential to establish a standardized anatomical reference point for measurement, ideally based on a specific vertebral level (e.g., the third lumbar vertebra, L3). This approach would ensure consistency across different patients and research centers, improving the validity and comparability of the findings with other studies. The authors are encouraged to explicitly define the vertebral level used for psoas muscle measurement and include this information in both the Methods section and relevant illustrative figures.

In accordance with this recommendation, we explicitly specified in the Introduction section (lines 65–66), the Materials and Methods section (lines 121), and Figure 1 that PMI, similar to SMI, is measured using CT or MRI images at the level of the third lumbar vertebra.

  1. The study is a single-center, retrospective analysis, which limits the generalizability of the findings; moreover, the lack of external validation for the PMI cut-off values weakens their clinical applicability. Authors should underline it in the conclusion.
  2. The optimal PMI cut-off values were derived statistically but not validated externally. Without broader validation, these values may lack general applicability in diverse patient populations.

We recognize this as one of the most significant limitations of our study and have explicitly stated in the Discussion section that “The PMI cut-off values identified in this study are specific to Japanese patients with HCC undergoing systemic targeted therapy and may not be generalizable to all patients with liver disease because skeletal muscle mass varies greatly depending on the stage of liver disease and race.” (lines 275–278) Furthermore, as a conclusion, we have added that these values may require revision based on findings from future prospective studies. (lines 297–298). We sincerely appreciate your invaluable feedback.

  1. The study appropriately uses time-dependent covariates (PMI, AFP, ALBI score), but it does not address potential biases from repeated measurements adequately.

As correctly noted, the frequency of PMI measurements and the intervals between them vary considerably across cases, resulting in numerous survival data points with exceedingly short observation periods. We have explicitly acknowledged this as a limitation of our study in the Discussion section (lines 284–286). Thank you for highlighting this critical point.

  1. The reported correlation between PMI and SMI is moderate (PCC = 0.38 for women which shows a weak correlation, PCC = 0.62 for men which shows a moderate correlation), particularly weaker in women, which challenges the claim that PMI is a reliable surrogate for SMI.

The slightly weaker correlation between PMI and SMI observed, particularly among women, may be attributed to factors such as the limited sample size, intra-observer variability, and the presence of survival data points with exceptionally short observation periods. This limitation has been addressed in the Discussion section (lines 288–291). We deeply appreciate your insightful feedback in identifying this pivotal issue.

  1. Sarcopenia is defined by both muscle mass and muscle strength. The study evaluates only mass, neglecting an essential diagnostic parameter, which reduces the robustness of the conclusions.

We recognize this as one of the limitations of our study and have explicitly stated in the Discussion section that “this study focused solely on skeletal muscle mass, without assessing muscle strength, which is essential for the diagnosis of sarcopenia. This retrospective study included many patients without grip strength measurements, precluding the evaluation of the prognostic impact of muscle strength.” (lines 278–282)

Minor Issues

  1. Complex sentence structures and redundancies, particularly in the Abstract and Introduction, reduce clarity. Some phrases, like "independently influenced survival following systemic targeted therapy," are repeated excessively.
  2. Key findings are unnecessarily repeated across multiple sections, reducing the manuscript's overall readability and impact.

In response to this suggestion, we have diligently removed redundant expressions and excessive repetitions throughout the manuscript. Thank you for bringing this important point to our attention.

  1. Figure 1: hight is not an English word, I think authors meant height as in the text. This must be corrected. Moreover, Height x Height can be represented in math as Height^2.

In accordance with this suggestion, we have appropriately revised Figure 1.

  1. Figure 3: please annotate median survival values and key survival time points on the Kaplan-Meier curves for enhanced interpretability.

In accordance with this suggestion, we have explicitly stated the 1-year survival rate and median survival time of the survival curve in Figure 3 within the manuscript (lines 189–192). Thank you for your valuable feedback.

Round 2

Reviewer 1 Report

Comments and Suggestions for Authors

The manuscript is OK now

Reviewer 2 Report

Comments and Suggestions for Authors

The revised manuscript has been much improved.

Reviewer 3 Report

Comments and Suggestions for Authors

No other modification required.